# Ultrasound Technologies and the Diagnosis of Giant Cell Arteritis

**DOI:** 10.3390/biomedicines9121801

**Published:** 2021-11-30

**Authors:** Dragoș Cătălin Jianu, Silviana Nina Jianu, Traian Flavius Dan, Georgiana Munteanu, Claudiu Dumitru Bîrdac, Andrei Gheorghe Marius Motoc, Any Docu Axelerad, Ligia Petrica, Anca Elena Gogu

**Affiliations:** 1Department of Neurosciences-Division of Neurology, Victor Babeș University of Medicine and Pharmacy, 300041 Timișoara, Romania; dcjianu@yahoo.com (D.C.J.); agogu@yahoo.com (A.E.G.); 2Centre for Cognitive Research in Neuropsychiatric Pathology (NeuroPsy-Cog), Department of Neurosciences, Victor Babeș University of Medicine and Pharmacy, 300041 Timișoara, Romania; claudiubirdac8@gmail.com (C.D.B.); amotoc@umft.ro (A.G.M.M.); docuaxy@yahoo.com (A.D.A.); ligia_petrica@yahoo.co.uk (L.P.); 3First Department of Neurology, Pius Brânzeu Emergency County Hospital, 300041 Timișoara, Romania; 4Department of Internal Medicine II, Centre for Molecular Research in Nephrology and Vascular Pathology, Victor Babeş University of Medicine and Pharmacy, 300041 Timișoara, Romania; 5Department of Ophthalmology, Dr. Victor Popescu Military Emergency Hospital, 300041 Timișoara, Romania; silvianajianu@yahoo.com; 6Department of Anatomy and Embryology, Victor Babeş University of Medicine and Pharmacy, 300041 Timișoara, Romania; 7Department of Neurology, General Medicine Faculty, Ovidius University, 900527 Constanța, Romania; 8Department of Internal Medicine II-Division of Nephrology, Victor Babeș University of Medicine and Pharmacy, 300041 Timișoara, Romania

**Keywords:** giant cell arteritis (GCA), “dark halo” sign, color Doppler imaging (CDI) of the orbital vessels

## Abstract

Giant cell arteritis (GCA) is a primary autoimmune vasculitis that specifically affects medium-sized extracranial arteries, like superficial temporal arteries (TAs). The most important data to be considered for the ultrasound (US) diagnosis of temporal arteritis are stenosis, acute occlusions and “dark halo” sign, which represent the edema of the vascular wall. The vessel wall thickening of large vessels in GCA can be recognized by the US, which has high sensitivity and is facile to use. Ocular complications of GCA are common and consist especially of anterior arterial ischemic optic neuropathies or central retinal artery occlusion with sudden, painless, and sharp loss of vision in the affected eye. Color Doppler imaging of the orbital vessels (showing low-end diastolic velocities and a high resistance index) is essential to quickly differentiate the mechanism of ocular involvement (arteritic versus non-arteritic), since the characteristics of TAs on US do not correspond with ocular involvement on GCA. GCA should be cured immediately with systemic corticosteroids to avoid further visual loss of the eyes.

## 1. Introduction-Giant Cell Arteritis (GCA)

Giant cell arteritis (GCA), also named temporal, granulomatous arteritis, or Horton disease, represents a primary autoimmune (non-necrotizing granulomatous) vasculitis [1,2,3,4,5,6].

It develops local activation of T cells and macrophages in the arterial wall and plays an important role in inflammatory cytokines, primarily IL1β, 6 and TNFα. Target antigens for the T-cell immune response are found in the inner elastic layer of the arterial wall [1,2,3,4,5,6].

GCA produces a segmental inflammation (discontinuous arterial involvement), with intermittent narrowing of the caliber of the artery, leading (by wall thickening) to partial obstruction (stenosis) or occlusion of the affected artery, its main clinical features being represented by signs of local ischemia [1,2,3,4,5,6]. The intensity of arterial injury is related to the proportion of elastic tissue in the media of the affected artery. On the one hand, because the intracranial arteries have less elastic fibers, they are rarely affected, resulting in fewer ischemic strokes. On the other hand, due to the fact that the extracranial arteries have more elastic fibers, GCA is of particular interest for extracranial medium-sized arteries (especially the superficial temporal arteries (TAs), or other branches of the external carotid arteries (ECAs)). Less often, it affects small extracranial arteries (like orbital vessels: posterior ciliary arteries (PCAs), or central retinal artery (CRA)), or large-sized arteries (aorta and its major branches) [1,2,3,4,5,6].

GCA is the most frequent sort of vasculitis that appears after the age of 50. Women are two to three times much more likely to be affected than men; the Caucasians, mainly northern Europeans and other patients in northern latitudes, are much more likely to be affected [1,2,3,4,5,6].

The American College of Rheumatology (ACR) modified criteria for the classification of GCA are as follows:An arteritis usually interesting the aorta and its major branches, especially the branches of the external carotid and vertebral arteries (the temporal artery being often affected);Usually patients with an age greater than 50 years at the appearance of clinical disease;Frequently associated with polymyalgia rheumatica, manifested by systemic symptoms, represented by fever, pain in the shoulders and hips, malaise, weight loss;New onset of a medium temporal headache;A clinically modified temporal artery (consisting in tenderness of the vessel or reduced temporal artery pulse), associated with scalp tenderness.Claudication of the jaw on mastication or tongue on mastication and on deglutition.An augmented erythrocyte sedimentation rate, more than 50 mm/h;A temporal artery biopsy (TAB) sample, indicating necrotizing of the vessel wall, with predominant mononuclear cell infiltrates or a granulomatous pathological process represented by multinucleated giant cells [7,8] (Figure 1) [9].

According to different authors, other suggestive features for GCA are a C reactive protein-PCR more than 1.5 mg/dL and frequent ophthalmological complications, consisting in acute, painless, and severe loss of vision in the affected eye [1,2,3,4,5,6].

The purpose of our review was to assess different imaging modalities (especially ultrasonography-US) that should be used to obtain the best diagnostic performance for the non-invasive diagnosis of GCA [9,10,11,12,13,14,15,16,17].

## 2. Introduction-Ultrasonography (US)

### 2.1. Ultrasonography Overview

Olah asserted that for US examination of extracranial arteries, distinct image modalities are important:Brightness mode (B-mode) Imaging. It uses a two-dimensional view of a portion of tissue because ultrasound signals are reflected from the target. The power of the reflected signal (named echo) is grayscale coded, being represented as a much or less bright dot, while the anatomical situation of distinct echoes depends on the depth of the insonated structure [18].The duplex image. It combines a B-mode image (which analyses the topography of the vessels) with pulse-wave (PW) Doppler (which focuses on flow velocities assessment). The two-dimensional B-mode view suggests that portion inside the vessel wherein a Doppler sample volume must be disposed and wherein the velocities are measured.

Linear segment array probes with frequencies ranging from 5 to 13 MHz should be used for medium-size arteries [18].
c.Color Doppler flow imaging. This technique measures the average frequency change in each sample volume. It associates a color flow map (representing the color-coded velocity information) with a B-mode image.

The first indicates the topography of the arteries, the sense of the flow (flow toward marked in red), and the localization of a turbulent flow or stenosis [18].
d.Power Doppler mode. It uses the strength of the return Doppler signal instead of the frequency shift. Signal strength is displayed as a color map superimposed on a B-mode image. Since Doppler power is determined primarily by the volume rather than the velocity of the moving blood, power Doppler imaging is free from aliasing artifacts and is much more sensitive to detect flow, especially in low-flow regions. Unfortunately, it does not include data on the direction or speed of the flow [18].


### 2.2. Systemic Arteries

#### 2.2.1. The Structure of the Wall of Systemic Arteries

According to different authors, all systemic arteries are composed of three histological layers: the intima, the media and the adventitia. A linear concentration of elastic fibers known as the inner elastic plate separates the intima from the media, and within the majority of the arteries, a second, less well-developed layer of connective tissue delimitates the adventitia from the media; this can be called the external elastic lamina [19,20]. Important variations in blood flow patterns and wall pathology are defined amongst arteries from distinctive vascular beds; the greater apparent example of various histological arterial characteristics is visible in the brain arteries. These arteries are exposed to low flow, they lack an external elastic lamina and their muscularis (i.e., media) appears thinner compared to different arteries. These arteries, unlike the rest of the vasculature, have a complicated collateral network exemplified with the aid of using the circle of Willis; however, these were also aided by at least seven different types of collateral flow from outside and inside the skull that confers its particular pathophysiology [19,20].

#### 2.2.2. Measures in the Arterial Wall

The preliminary levels of atherosclerosis, which represents a persistent inflammatory process, consist of intima thickening that during a few instances can regress and normalize. However, it can also progress and turn pathological while greater fibrotic tissue is incorporated [21,22,23].

In 2004, an international group of researchers with experience in US met in Manheim, Germany, with the goal of standardizing the protocols for carotid wall imaging and facilitating comparisons across studies [24]. The two most typical measures within the carotid artery wall are the carotid intima-media thickness (cIMT) and carotid plaque (CP) [24,25,26,27,28,29].

#### 2.2.3. Carotid Intima-Media Thickness (cIMT)

It represents the distance between the luminal–intimal interface and the media-adventitial interface of the carotid arteries. This is a double-line pattern displayed by US in brightness mode (B-mode) on the wall of the carotid artery in a longitudinal plane [24,25,26,27,28,29]. The cIMT varies with age, sex, race-ethnicity and body surface area. The average cIMT in population-based studies increases from 0.5 mm in young individuals to 0.80 mm in the elderly, and, for the age group usually evaluated in stroke prevention, it is considered normal below 1.0 mm, and increased between 1.0 and 1.4 mm. For values equal to 1.5 mm or higher, the parallel course of intima-media and adventitia layers is often lost, and the IMT turns into a plaque [24,25,26,27,28,29].

According to Del Sette, cIMT may represent pathological changes in medial arterial hypertrophy or thickening of the intima in the absence of atherosclerosis, like in GCA, in which case cIMT does not represent atherosclerosis [25].

#### 2.2.4. Carotid Plaque (CP)

Is defined as a focal extrusion into the arterial lumen of at least 50% more in thickness than the surrounding cIMT value, or an absolute thickness greater than 1.5 mm when measured in the same fashion as cIMT [25,26,27,28,29].

### 2.3. Advantages of Ultrasonography (US)

On the one hand, US is successful to investigate vessel wall anatomy and to identify different parietal anomalies (wall thickening in the presence of atherosclerosis or vasculitis, like in GCA, hypoechoic plaques, clotting, parietal hematoma, and dissections), and the outside diameter of the artery; it could rule out each stenosis and occlusion, particularly in the carotid bulb [24,25,26,27,28,29]. On the other hand, the usage of a color-coded Doppler flow imaging method improves the study of arterial and venous blood flow characteristics, which performs a crucial position in the distribution of atherosclerotic plaques and provides information on smaller arteries with low flow [24,25,26,27,28,29].

## 3. Ultrasonography (US) in Giant Cell Arteritis (GCA)

### 3.1. Background

The advantages of US over different imaging techniques in GCA are described by its swiftness (approximately 15–20 min, if it’s executed by a skilled sonographer) and its high resolution (a high–frequency probe offers each an axial and a lateral resolution of 0.1 mm in B-mode) [30,31,32,33,34,35,36,37]. In addition, US is more sensitive than TAB, because the later exams only have a limited anatomical location in systemic disease, like GCA.

Due to the fact that GCA can affect multiple large, medium, and small size extracranial arteries, we have to assess the TAs by US, as well as:Other medium-sized arteries (branches of the ECAs): the internal maxillary artery (claudication of the jaw on mastication), the renine artery (claudication of the tongue on mastication or on deglutition), the facial, and the occipital arteries,Large size arteries: the common carotid arteries (CCAs), the ECAs, the internal carotid arteries (ICA’s), the vertebral, the subclavian, and the axillary arteries,Small size arteries: the ophthalmic arteries (OAs) and the CRAs and PCAs [9,10,11,12,13,14,15,16,17].

### 3.2. Ultrasonography (US) of the Temporal Arteries (TAs) and Other Medium Size Arteries

The common superficial TA divides into the frontal and parietal ramus, just immediately before the ear. The distal part of TA and their two rami are localized between the two layers of the temporal fascia, which can be identified by US exam as a brilliant band. Both the lumen size of the TA and both layers of this fascia, including the TA’s wall, measure 0.7 mm, respectively [31,32,33,34].

#### 3.2.1. Technical Requirements (According to Schmidt)

Linear transducers must be used with a minimum grayscale frequency of 8 MHz. Color frequency should be about 10 MHz [31,32,33,34].

#### 3.2.2. Machine Adjustments (According to Schmidt)

The pulse repetition frequency (PRF) should be 2.5 kHz as peak systolic velocities are rather high (20–100 cm/s) [31,32,33,34].

#### 3.2.3. Sequence of the US Exam (According to Schmidt)

The color Doppler US exam includes eight segments of the TAs in two planes (longitudinal and transversal scans): common superficial TA, parietal proximal, parietal distal (>2 cm distal from the bifurcation) frontal rami on both sides. If the color signals observe localized aliasing and persistent diastolic arterial flow, one should utilize the power-Doppler mode to identify the stenoses [31,32,33,34].

Ultrasound exam observes that inflammation is often segmental, due to a discontinuous vessel damage-skip lesions in temporal arteries or other branches of ECAs [31,32,33,34,35,36,37,38].

There are four main findings noted in the US diagnosis of temporal arteritis:*“Dark halo” sign:* An usually homogeneous, hypoechoic wall thickening surrounding the lumen of an inflamed artery. It is well outlined towards the luminal side, visible both in longitudinal and transverse planes, most commonly concentric in transverse.

It describes the edematous arterial wall swelling, whereas histology exam displays cell infiltrates and granulomas. TAB may miss the pathological zone because of the segmental appearance of temporal arteritis [30,31,32,33,34,35,36,37,38,39,40,41] (Figure 2) [11].

b.*Stenoses* are characterized by aliasing and persistent diastolic flow by colour Doppler US. The peak systolic velocity (PSV) assessed within the stenosis area by pulsed-wave Doppler US is two or more times greater than the PSV recorded in the prestenotic segment of the vessel, with turbulence at the level of stenosis, associated with diminished velocities distal to the stenosis [30,31,32,33,34,35,36,37,38,39,40,41] (Figure 3) [11].

c.*Acute occlusions,* wherein the US image is similar to that of acute embolism in different other vessels, with lack of color Doppler signals (even with low pulse repetition frequency and high color gain) in a visible artery lumen filled with hypoechoic material (cloth) [30,31,32,33,34,35,36,37,38,39,40,41].d.*Compression sign.* The thickened vessel wall remains visible upon compression by the ultrasound examiner; the wall swelling is hypoechogenic (in acute temporal arteritis), contrasting with the mid-echogenic to hyperechogenic surrounding tissue [38].

Similar US patterns can be found in other medium-size arteries: the inner maxillary, the facial, the lingual, and the occipital arteries [9,10,11,12,13,14,15,16,17].

US investigation should be executed before the beginning of the corticoids therapy, or in the first seven days of treatment, because with corticosteroids the “halo” showed by TAs US withdraws in 2–3 weeks. Nevertheless, the diagnosis process should not delay the introduction of corticosteroids. US may also detect inflamed TAs in patients with clinically normal TAs, but with clinical signs of polymyalgia rheumatica [30,31,32,33,34,35,36,37].

In a meta-analysis of studies on temporal artery ultrasound Karassa et al. noted a sensitivity of 87% and a specificity of 96% for the clinical diagnosis of temporal arteritis [41].

Concentric hypo-echogenic mural thickening or dubbed “halo” is considered by Schmidt to be the most specific (99.5% specificity) and sensitive (72% sensitivity) sign for GCA, representing the vessel wall edema” [31,32,33,34,40].

In 2010, Arida et al. assessed in a meta-analysis the specificity and the sensitivity of the “halo” sign identified by TAs US for GCA diagnosis versus the American College of Rheumatology (ACR) 1990 criteria for the classification of GCA (utilized as a reference standard) [37]. Only eight research comprising 575 patients, 204 of whom have been ultimately recognized with GCA, met the technical quality standards for US. The halo sign presented a sensitivity of 68% and a specificity of 91% (when it was unilateral), and a sensitivity/specificity of 43% and 100%, respectively (when it was bilateral) in TAs US exam for GCA diagnosis when the 1990 ACR criteria represented the reference standard [37]. Arida et al. noted that the halo sign detected in US exam is very important for the diagnosis of temporal arteritis, indicating vasculitic wall edema, in GCA [37].

According to different authors [37,42,43], now, TAB is no longer beneficial and sustained if there are significant clinical and sonographic findings for temporal arteritis.

Practically, US replaced TAB as the main investigational method in the GCA patients, because it has some advantages:The results of TAB, which is an invasive method, appear only after a few days, sometimes with inconclusive results, because skip lesions have been noted in 8.5% of the biopsies in GCA cases Biopsy may miss the lesion because of the segmental appearance of GCA [42].US can assess the whole length of the temporal artery and other branches of ECAs, and large supra-aortic arteries (like CCAs and axillary arteries) [43].US is a non-invasive procedure and can be realized as a complementary method to the clinical assessment without any delay [43].

### 3.3. Duplex and Color-Coded Duplex Sonography of the Large Cervical and Cervico-Brachial Vessels

The Chapel Hill Consensus Conference (2012) considered large vessel vasculitis (LVV) as vasculitis affecting the aorta and its major branches more often than other type of vasculitides; however, any size (large, medium, small) of an artery may be affected [8,30].

For example, in GCA, could be affected at the same time: (a) large arteries (e.g., aorta, the subclavian and axillary arteries, the CCAs, the ICAs), (b) medium arteries (e.g., TAs, inner maxillary arteries), and small arteries vascularizing the eye and orbit (e.g., CRA, or PCAs) [8,30,31,32,33,34,35,36,37,38,39].

LVV GCA has been previously disregarded and underdiagnosed. However, there is important evidence confirming that large arteries are affected in around two-thirds of GCA cases and one-third of patients with polymyalgia rheumatica (PMR) [39].

Sturzenegger asserted that angiography could not illustrate the vessel wall anatomy. Consequently, for diagnosing inflammation of the cervical and cervico-brachial large vessels, US can be very helpful, as it can identify changes of the vessel walls, like dark halo sign (by using B-mode imaging) and it can assess arterial stenosis or occlusions (with pulse-wave-PW Doppler flow velocities measurements, and Color Doppler Duplex sonography) [30].

According to different authors, there are two US features of large vessels GCA:Vessel wall thickening, represented by the dark halo sign, which is homogeneous, circumferential and overlong segments [8,30]. According to Diamantopoulos and al, the cut-off limit for vasculitis (GCA) for the CCAs is 1.5 mm and for the axillary arteries is 1 mm [43].Stenosis, due to a segmental inflammation, which produces a discontinuous arterial involvement (hourglass-like) [8,30].

The arterial wall inflammation, stenosis, or occlusions of the large arteries (e.g., CCA, ICA) persists for months, despite corticosteroid treatment [8,30] (Figure 4 and Figure 5) [10].

According to Schmidt et al., US exam of axillary arteries represents a useful noninvasive technique to identify large-vessel GCA that may appear with or without temporal arteritis [44].

Sturzenegger noted that differential diagnosis of large vessels GCA with atherosclerosis is essential in cases aged over 50, given that GCAs affecting the large vessels almost exclusively impair this category of patients. There are several characteristics of an arteriosclerotic wall: the thickening usually is less homogeneous; we can identify calcified arteriosclerotic plaques ulcers; stenosis is noted over shorter segments, they are not concentric, not tapering, and the site of the lesion is different (e.g., mainly bifurcations: carotid bifurcation, etc.) [30].

According to Sturzenegger, differential diagnosis of LVV GCA with others LVV, especially Takayasu arteritis, has to be realized:Takayasu arteritis especially affects young women (below 40 s);Tender scalp or polymyalgia syndrome are very rare in Takayasu arteritis.The involvement of CCA is more frequent in Takayasu arteritis, while temporal arteries are not affected in Takayasu arteritis.US image of wall thickening (“halo”) is brighter in Takayasu arteritis than in GCA, because the patient with GCA has a larger mural edema than in Takayasu arteritis (GCA being a more acute disease than Takayasu arteritis) [30].

### 3.4. Color Doppler Imaging (CDI) of Orbital (Retro-Bulbar) Vessels

Permanent visual loss has been reported to occur in up to 19%, and visual symptoms in up to 31% of acute GCA cases [40]. Unfortunately, 20% of GCA cases with visual loss have occult GCA, without systemic manifestations [45].

This is why early diagnosis and therapy with glucocorticosteroids protect against visual loss [46].

However, if the visual loss has already appeared therapy with glucocorticosteroids is ineffective [47].

For all these reasons, Diamantopoulos et al. examined the fast-track outpatient GCA clinic (FTC), based on quick clinical, laboratory and US evaluation (scanning in maximum 24 h after clinical exam of just temporal, axillary, and carotid arteries) of the cases suspected to have GCA and immediate therapy if appropriate. The main objective of their study was to assess whether the rate of visual loss in GCA cases was lower in the period with the FTC approach compared with the period before, with the conventional exam. They concluded that the implementation of the FTC in GCA management appeared to significantly decrease the risk of permanent visual loss [43].

In conclusion, a significant percentage of patients with GCA detected by TAB have ophthalmological complications, clinically manifested by unilateral sudden, painless, and sharp loss of vision due to vasculitic involvement of the small retrobulbar arteries in the affected eye [48,49,50,51,52,53,54,55,56,57]:Arteritic Anterior Ischemic Optic Neuropathies (AAION) results from short posterior ciliary arteries (PCAs) vasculitis and the consecutive optic nerve head (ONH) infarction [48,49,50,51,52,53,54,55,56,57], or,Central Retinal Artery Obstruction (CRAO) occurs when the thrombotic blockage produced by the vasculitic process due to GCA is within the optic nerve substance [48,49,50,51,52,53,54,55,56,57],

Other ophthalmological complications are represented by branch retinal artery occlusion (where arterial branches that supply the inner layer of the retina are affected; their occlusion leading to a sectoral pattern of retinal opacification), diplopia (which is most commonly caused by abducens nerve palsy) and amaurosis fugax (which is a transient monocular vision loss) [48,49,50,51,52,53,54,55,56,57].

According to Schmidt et al., among different patients with acute temporal arteritis and concomitant visual symptoms, unlike for TAB, there was no correlation between the findings of TAs US and the occurrence and severity of eye involvement in newly diagnosed, active GCA. US identifies edematous wall swelling, whereas histology displays cell infiltrates and granulomas. Visual complications appeared less frequently if proximal arm large-vessel GCA was present. (axillary arteries were affected) [40].

For this reason, and because ophthalmological complications are frequent in GCA, we always have to exam by duplex ultrasonography the orbital (retro-bulbar) vessels in patients with known GCA or in cases of unilateral, acute, painless, and severe loss of vision [9,10,11,12,13,14,15,16,17].

#### 3.4.1. The (Intra) Orbital (Retrobulbar) Arteries

The orbit is vascularized, especially by the ICA (via the ophthalmic artery-OA and its rami) with a reduce contribution from the ECA. The vessels of the eye and orbit present an inter-individual variation.

The OA represents the collateral rami of the ICA, which vascularizes the eyeball and its muscles and the orbit and surrounding parts. It branches in several collateral arteries (central retinal artery (CRA), posterior ciliary arteries (PCAs, etc)) and finishes near the superomedial orbital margin in two terminal rami: the dorsal nasal artery and the supratrochlear artery, both supplying the scalp and the forehead. The ECA gives two branches to the orbit (the infraorbital artery and an orbital branch from the middle meningeal artery) [48,49,50,51,52,53,54,55,56,57].

CRA (Figure 6A), (Table 1) [11] goes forward in the dura mater inferior to the optic nerve (ON), and about 1–1.5 cm behind the eyeball penetrates the ON, entering the globe to feed the inner retinal layers (these terminal branches being the only blood source to the greater part of the retina). The fovea and a small zone around it are not vascularized by CRA or its terminal rami, but instead by the choroid. In 1/5 of the population, the area of retina situated between the macula and the ON (including the nerve fibers from the photoreceptors situated in the fovea) is supplied by the cilio-retinal artery (a rami of the ciliary circulation). In these cases, the central vision will be preserved in the situation of central retinal artery occlusion (CRAO) [48,49,50,51,52,53,54,55,56,57].

PCAs (Figure 6A), (Table 1) [11] are 1–5 collateral rami of OA, that run forward and subsequently divide into multiple rami (the medial and lateral long PCA and multiple short posterior ciliary arteries), which penetrate the sclera posteriorly near the ON and macula to vascularize the outer coats of the globe [48,49,50,51,52,53,54,55,56,57]. According to Hayreh, the PCAs are end arteries (no anastomose with any other artery), thus, any acute occlusion of a short or long PCA will determine a diminished choroidal infarct, belonging to the greater zone vascularized by the PCAs [53]. The long PCAs, two in each eye, vascularize the iris, ciliary body and choroid. The short PCAs, approximatively 20 rami, emerge from the medial (nasal) PCA and lateral (temporal) PCA and vascularize the choroid and ciliary processes, and the optic disc (via the circle of Zinn, which is an anastomotic arterial ring) [40,41,42,43,44,45,46,47,48,49]. The arterial flow of the orbit is autoregulated by the autonomic nervous system and chemical mediators, like adenosine, nitric oxide, or endothelin 1 [48,49,50,51,52,53,54,55,56,57].

The orbital venous system is represented by two constant veins, the superior ophthalmic vein (SOV) and the inferior ophthalmic vein (IOV), and multiple variable veins [48,49,50,51,52,53,54,55,56,57].

##### Probe Selection

US devices equipped with a linear probe at 6–12 MHz (up to 15 MHz) are suitable for detection at a depth of 1.4 to 4 cm (with color Doppler US, as a guide) and for quantitative assessment (with pulsed Doppler spectral analysis) of blood flow in orbital vessels [12,16,58,59,60].

##### Technique

Color Doppler US is used with the gain modified to avoid artifacts, thus permitting identification of low velocities. The Doppler sample gate (1.5 mm) is then applied to the localized vessel to assess velocities. An angle correction between 0–60 o should be utilised when the retrobulbar vessels are not parallel to the US beam [12,58,59,60].

Imaging of the OA: The OA is identified deeper in the internal orbit, at a depth of 35 to 36 mm, medial to the ON and 1.5 mm posterior to the eye. It presents a diameter of 0.7 to 1.5 mm. It has a forward red-coded blood flow. It has a moderate to high RI, a prominent initial systolic peak with a typical incisura, and much less diastolic flow; its arterial waveform suggests a peripherally situated high-resistance artery [12,58,59,60].

Imaging of the CRA: The CRA is identified just below the optic disc (<1 cm), and has a forward red-coded blood flow. It presents an estimated diameter of 0.2 mm. It shows a reduced RI, low PSV, and a rapid flow in the diastolic phase (a suggestive low resistance waveform with the continuous diastolic flow) [12,58,59,60].

Imaging of the central retinal vein (CRV): It is a small-caliber vessel that runs parallel to the CRA, with a continuous venous flow in the opposite direction of CRA (with flow waveform below the zero lines) [12,58,59,60]. (Figure 6A, Table 1) [11].

The imaging of the CRA and CRV is observed inside the ON, a few millimeters posterior to the lamina cribrosa (see Figure 6A).

Imaging of the short posterior ciliary arteries (sPCAs): The sPCAs consist of numerous rami divided into two groups (the nasal and temporal PCAs) which are identified along both sides of the ON at 1 to 3 mm behind the posterior wall of the eye. The arteries have a forward red-coded blood flow. They present a diameter of 0.1 to 0.2 mm. Their flow is a low RI with velocities between the OA and the CRA velocities [12,58,59,60]. (Figure 6B, Table 1) [11].

Imaging of the SOV: The SOV is situated in the superonasal orbit. Due to the lack of valves and their hypotonia, the venous flow waveform varies in function of different external factors [12,58,59,60].

Although the US equipment can measure the vessel diameter, the caliber of the orbital vessels is too small to estimate their diameter [12,58,59,60].

Several Doppler Indices can be extracted from the velocity wave: [12,58,59,60].

Peak Systolic Velocity (PSV): The highest velocity observed during the systole of the Doppler waveform.End Diastolic Velocity (EDV): The lowest velocity noted during the diastole of the Doppler waveform.Resistivity Index (RI): PSV-EDV)/PSV.Mean Flow Velocity (MFV)Pulsatility Index (PI): PSV-EDV)/MFV.

Documentation of the exam contains information about the probe, power, eye (right, left), velocities, RI, PI and interpretation of results: alteration of velocities, RI and Spectral PW Doppler. These abnormal aspects are due to either local pathology (like stenosis or occlusion), or are the result of upstream hemodynamic alterations (due to ipsilateral ICA stenosis and insufficiency of collaterals at the level of the polygon of Willis) [12,58,59,60].

##### Arterial Blood Supply of the Optic Nerve Head (ONH)

Depending on the arterial blood supply, ON is divided into anterior (the optic nerve head-ONH) and posterior regions.

The ONH contains four regions (from ventral to dorsal):the surface nerve fiber layer,the prelaminar region,the region of the lamina cribrosa, andthe retrolaminar region [11,50,51,52,53,54].

The PCAs are the main source of arterial supply for the ONH, via the peripapillary coroid and the short PCAs (or, sometimes, via the circle of Zinn and Haller). Due to this sectoral arterial supply in the ONH, the segmental visual loss in anterior ischemic neuropathy could be explained [11,50,51,52,53,54].

##### Pathophysiology of Factors Controlling Blood Flow in the ONH

The arterial flow in the ONH is influenced by three factors: the resistance to blood flow, the arterial blood pressure (BP), and the intraocular pressure (IOP):

Perfusion pressure = Mean B—intraocular pressure (IOP), where: Mean BP = Diastolic BP + 1/3 (systolic BP—diastolic BP) [11,50,51,52,53,54].

The resistance to blood flow is determined by the diameter of arteries that vascularize the ONH; this caliber depends on the following factors: the efficiency of self-regulation of the ONH arterial flow, the modifications in the arteries irrigating the head of the ON, and the rheological features of the blood, especially viscosity [11,50,51,52,53,54]The arterial blood pressure (BP).

Both arterial hyper- or hypotension can disturb the arterial flow of ONH in distinct modalities.

In the former case, the most important mechanisms are represented by an augmented arterial resistance in the terminal arterioles, consecutive hypertensive modifications in the arteries of ONH, and disturbances of the autoregulation of the arterial flow.

In the latter case (of arterial hypotension), a severe diminution of BP below a critical level of self-regulation, would decrease the arterial flow in the head of the ON. The main causes consist in systemic hypotension (arterial hypotension during nocturnal sleep, intensive antihypertensive drugs, etc.,) or a local ocular/ONH arterial hypotension (due to stenosis of the implicated arteries, such as internal carotid, ophthalmic or one or more of the PCAs (vascularizing the ONH), or a drop of perfusion pressure in the neighboring peripapillary choroid [11,50,51,52,53,54].

c.The intra-ocular pressure (IOP)

There is an opposite relationship between IOP and perfusion pressure in the ONH, due to the fact that the arterial flow in the head of the ON depends upon the perfusion pressure, which is equal to mean BP minus IOP. Thus, a greater augmentation in IOP would be needed to influence significantly the ONH arterial flow in healthy subjects (with normal BP and autoregulation) than in patients with arterial hypotension, deregulated autoregulation [11,50,51,52,53,54].

#### 3.4.2. Anterior Ischemic Optic Neuropathies (AIONs)

AIONs consist of a segmental infarction of the ONH vascularized by the PCAs. Diminished/absent arterial supply can appear with or without inflammation of these arteries [11,50,51,52,53,54].

Arteritic AION (A-AION) results especially from PCAs trunk arteritis, which produces an inflammatory thickening of the vessel wall, +/− stenosis, and, sometimes, intraluminal thrombi with occlusion of the artery [11,50,51,52,53,54].

Nonarteritic AIONs (NA-AIONs) are produced by transient nonperfusion or critical hypoperfusion of the supplying arterioles of the ONH (para-optic branches of the PCAs). NA-AIONs are a multifactorial disease, the nocturnal arterial hypotension (which is a systemic cause) being the most important risk factor. Often, NA-AION patients have a loco-regional cause (especially an anatomical predisposition, consisting in a small disc, with crowding of ONH), which may combine with hypoperfusion of the ONH. Less frequently, the nonarteritic form of AION is due to emboli in the arteries/arterioles supplying the ONH [11,50,51,52,53,54].

Anterior segment examination of both eyes is generally normal in all AION cases (arteritic or nonarteritic)

The patients with A-AION present a monocular, acute, painless, permanent severe deterioration/loss of vision, sometimes preceded by amaurosis fugax, associated with concomitant diffuse pale optic disc edema [11,50,51,52,53,54].

On the contrary, the patients with NA-AION have less permanent deterioration of vision, which are never preceded by amaurosis fugax, usually associated with a concomitant hyperemic optic disc edema. The optic disc in the contralateral eye frequently is small in diameter with a small/absent physiological cup (disc at risk) [11,50,51,52,53,54].

##### Color Doppler Ultrasonography of Intraorbital Arteries in A-AIONs

In the acute phase of unilateral clinical eye involvement, absent (undetectable) signals in the homolateral PCAs (not corresponding to homolateral internal carotid artery occlusive disease) are classified as Doppler US features in acute arteritic AION (consecutive to GCA). In addition, we can identify a high resistance index (RI), with decreased velocities (especially EDV) in all retrobulbar vessels, in both orbits [9,10,11,12,13,14,15,60,61,62] (Table 2) [13].GCA acute cases with no evident clinical ocular involvement present a decrease in arterial flow in bilateral orbits, with increased RI, and diminished velocities (especially EDV). The severely diminished flow in the PCA, associated with diminished flow in the CRA and very high flow in the OA (all on the affected side) are the common US features in this type of patient. This US aspect is an essential predictor of an imminent A-AION and needs prompt treatment with high-dose corticosteroids [9,10,11,12,13,14,15,60,61,62] (Figure 7) [11].

According to Jianu et al., [15], the analysis of the data (Table 2) indicated that a threshold value of 0.71 for the RI of the temporal PCAs in the clinically affected eye with A-AION, realized the best association of sensitivity (Se) (86%), Specificity (Sp) (96%), positive predictive value (PPV) (88%), and negative predictive value (NPV) (96%), respectively. A threshold value of 0.68 for the RI of the nasal PCAs in the clinically affected eye, realized the best association of Se (86%), Sp (93%), PPV (76%), and NPV (96%), respectively.

##### Color Doppler Ultrasonography of Intraorbital Arteries in NA-AIONs

Velocities and RI in PCAs are generally preserved. We observe only a slight diminution of PSV in PCA in the clinically affected eye, with a very low decrease of PSV in homolateral CRA, due to associate papillary edema [9,10,11,12,13,14,15,60,61,62].

In OAs, PSVs could be variable (between normal and diminished, in function of the homolateral ICA status). For example, a severe ICA stenosis (≥70% reduction of vessel diameter), associated with an insufficient Willis polygon, leads to diminished PSV in homolateral OA. Homolateral ICA severe stenosis/occlusion can contribute to the development of NA-AION either by embolism or by transient non perfusion or critical hypo perfusion of paraoptic branches of PCAs that supply the ONH [9,10,11,12,13,14,15,60,61,62].

Hayreh asserted that the embolic occlusion of the trunk of the PCAs or of the ONH arterioles (paraoptic branches of PCAs) appears much less frequently than thrombotic occlusion (especially due to GCA). Embolic cause of NA-AION can be suspected if the patients have: (a) sudden onset of visual loss, not related to sleep or any other factor associated with arterial hypotension; (b) the optic disc has a normal cup; (c) aspect of obstruction of a PCA on fluorescein fundus angiography, and on Color Doppler US of orbital arteries, but (d) no clinical features suggestive of GCA [13,52,53]

#### 3.4.3. Central Retinal Artery Occlusion (CRAO)

An abrupt severe decrease of arterial blood flow through the CRA generates CRAO, producing critical ischemia of the inner retina. CRAO determines acute, painless, and severe loss of vision in that eye, because there are no functional anastomoses between the CRA (that vascularizes the retina), and the PCAs (that supply the coroida) [11,16,17,55,56,57].

CRAO is characterized by the following key features:Sudden, painless, and sharp loss of vision of the affected eye [11,16,17,55,56,57].Anterior segment examination is normal in both eyes [11,16,17,55,56,57].On funduscopy, the entire retina, with the exception of the fovea, appears pale, swollen and opaque (ischemic whitening of the retina as a result of cloudy swelling), while the central fovea still looks reddish (the so-called “cherry-red spot” in the middle of the retina). The central fovea is supplied by the cilio retinal artery, which is a rami of the short PCA. Thus, in CRAO, there is a relatively intact choroidal circulation (blood flow in PCA is normal), in contrast to the ischemic retina (where CRA is obstructed) [11,16,17,55,56,57].

Frequently (75–80%), the blockage is situated within the ON, thus explaining why the cause of this blockage is not visible on ophthalmoscopy. The majority of CRAO’s are generated by thrombus appearance in situ due to systemic diseases: hematological diseases, or systemic vasculitis (especially GCA). This is a solid argument for a systemic assessment of all CRAO patients [11,16,17,55,56,57,60,61,62].

In only a quarter of cases are emboli detected by ophthalmoscopy in the CRA, suggesting that an embolic etiology is less frequent in CRAO (cardio or arterial emboli) [11,16,17,55,56,57,60,61,62]

##### Color Doppler Ultrasonography in the CRAO

In acute unilateral CRAO, Color Doppler US of the retrobulbar vessels observes a typical finding, represented by a severely diminished or absent blood flow in the CRA of the clinically affected eye (with absent or severe decrease velocities, especially EDV, and high RI). Generally, this type of patient presents normal flow in the PCAs (the choroidal branches of OA) and OA [11,16,17,55,56,57,60,61,62] (Figure 8A–D) [11]. In contrast, in rare patients with ocular ischemic syndrome (OIS), absent or diminished flow in CRA is associated with severe diminished/absent flow in the PCAs. In addition, OIS presents an absent or reversed end-diastolic flow in the OA and even the CRA. This “steal sign” is characteristic of severe stenosis or occlusion of the homolateral ICA, with an inefficient circle of Willis and of the sphenoidal artery (accessory ophthalmic artery) [11,16,17,55,56,57,60,61,62].

Furthermore, this US technique can identify calcic (hyper-echoic) emboli inside the CRA, *(retrobulbar “white spot sign”)*. This US feature presents an excellent interobserver agreement, rules out a GCA etiology of CRAO, and confirms the embolic cause of CRAO, but does not clearly identify the etiology of the embolus (cardiac, arterio-arterial) [63].

## 4. US and Others Imaging Techniques


**a. For the assessment of large arteries (aorta and its major supra-aortic branches), and medium arteries (for example, TAs and other branches of ECAs)**


According to the most recent European Alliance of Associations for Rheumatology (EULAR) guidelines, a suspected clinical diagnosis of GCA should be confirmed by different imaging techniques (US or MRI for temporal or other medium cranial arteries), US, CT, positron emission tomography-computed tomography (PET-CT) or MRI for the aorta/large extracranial arteries, or TAB (with histopathological exam) [64]. More, EULAR guidelines recommend performing a second type of test if the first one is negative, but the clinical suspicion of GCA persists [64].

*Contrast-enhanced, high-resolution MR imaging* realizes a noninvasive exam of the inflammation of the arterial wall of the temporal arteries, but there is little expertise in large arteries assessment. The significant features are represented by circumferential mural thickening and wall edema (the last one is enhanced in T2 sequences). Because MRI does not need iodinated contrast or ionising radiation, it has been utilized for repetitive exams [39,65,66].

According to Blay, the differences between the diagnostic power of high-resolution MRI and color-coded duplex US of extracranial arteries in detecting GCA are not significant (Sensitivity of high-resolution MRI and color-coded duplex US was 69% and 67%, respectively, while specificity was 91% in both). Thus, both techniques may be useful in the exam of cases with suspected GCA [65].

Lecler et al. conducted a study whose main purpose was to identify which combination of imaging modalities should be used to obtain the best diagnostic performance for the non-invasive diagnosis of GCA [66]. Patients underwent high-resolution 3 T magnetic resonance imaging (MRI), temporal and extracranial arteries US and retinal angiography (RA), prior to TAB. The diagnostic accuracy of each imaging modality alone, then a combination of several imaging modalities, was evaluated. The diagnostic algorithm with the overall best diagnostic performance was the one starting with MRI, followed either by US or RA, yielding 100% sensitivity (22/22; 95% CI: 85−100%), 100% (15/15; 95% CI:78−100) and 100% accuracy (37/37; 95% CI: 91−100) [66]. They concluded that the use of MRI as the first imaging examination followed by either US or RA reaches high degrees of performance for the diagnosis of GCA and is recommended in daily practice [66].

*Computed Tomography (CT) and Computed tomodensitometric angiography (CT-A)* exam (with a short scanning time) the arterial wall and endoluminal part of large arteries (including the thoracic and abdominal aorta) [39]. Prospective GCA patients assessed by CTA have revealed LV involvement in 45–68% of subjects [67,68,69]

Typical signs of large arteries vasculitis are circumferential wall thickening and vessel wall contrast enhancement. However, CTA features may be affected by a period of only three-day corticosteroid therapy [67].

The concomitant exam of aortic aneurism and the accurate differential diagnosis between arteritis (concentric thickening of the wall) and atherosclerosis (eccentric thickening of the wall, with focal calcifications, are important advantages of CTA. The disadvantages of these techniques are invasiveness, nephrotoxicity, and high radiation exposure when repetitive exams are realized, but recent low-dose CTA techniques may decrease radiation exposure [10,39,69].

*18 Fluorodeoxyglucose-**positron emission tomography (18 FDG-PET)* may identify with accuracy large vessel vasculitis, such as GCA (because it detects inflamed walls that have an augmented glucose metabolism with an increased vascular 18FDG uptake). According to different 18FDG-PET studies, 58–83% of GCA cases present large vessels involvement [39,70,71,72].

This method realizes with great accuracy the differential diagnoses with infectious or neoplastic disease, but it is not as accurate in identifying stenosis or occlusions and differentiation of the vasculitic lesions with the atherosclerotic plaques that also presents augmented vascular uptake (especially in elderlies) [10,39]. Another inconvenience is represented by the fact that a consensus agreement concerning 18FDGPET criteria of Large vessels vasculitis is missing. 18FDG uptake equal to or greater than liver uptake on PET has been considered as the best criterion of large vessels GCA [73].

The vascular enhancement in large vessels GCA is also diminished after three-day of corticosteroid therapy but has adequate sensitivity for diagnostic objectives [10,39].

Few studies associated ultrasound and PET-CT, to diagnose GCA, instead of assessing each tool separately for diagnostic utility [74,75]. They concluded that PET/CT measuring vessel wall metabolism and US vessel wall morphology showed a comparable diagnostic accuracy for GCA. Because PET/CT and US presented different results within single vascular regions, they should be used as complementary methods, with a second imaging technique augmenting the diagnostic yield by 16–20% [74,75,76].

18FDGPET/CT cannot identify inflammation in the temporal arteries; therefore, it is not suitable for the diagnosis of temporal arteritis (the exam of medium-size arteries, especially rami of the ECA) and cannot replace US, *Contrast-enhanced, high-resolution MR imaging* MRI or temporal artery biopsy [11,39,65].

**b. For the assessment of small retro-orbital arteries (CRA, PCAs, OA)** in patients with GCA with eye involvement, we can utilize two types of techniques: an invasive one (the fluorescein angiography/retinal angiography (RA)) and diverse noninvasive imaging methods, such as Color Doppler ultrasonography of the retrobulbar vessels, structural Optical Coherence Tomography (OCT) of the ONH and OCT angiography [9,10,11,12,13,14,15,50,66,76].

Fluorescein angiographic data reveal a characteristic sign, which is represented by a severely delayed and extremely poor or absent filling of the optic disc and choroid in arteritic AION (A AION); this sign differentiates the arteritic form from the nonarteritic form of AION (NA AION), where fluorescein angiographic exam observes only impaired optic dis supply. Severe delayed optic disc filling represents a sign of ischemia, which is not noted in optic disc edema produced by papilledema or other nonischemic etiologies [50,51,52,53,54]. Additional data are provided by Color Doppler US of orbitals vessels, which are represented by absent flow signal or high RI in the affected sPCAs in A AIONs, in contrast with relatively preserved velocities and RI in sPCAs in NA-AION [60,61,62].

Both methods confirm the histopathological exam, that certify vasculitis of the whole trunk of the sPCAs in A AION (with consecutive nonperfusion at choroidal and optic disc levels, and the subsequent ONH infarct). In NA AION, the affected flow to the ONH is distal to the sPCAs, probably affecting only the para-optic rami of sPCAs, that vascularize only the ONH (affected optic disc vascularization, with normal supply of the choroid). The insufficiency of the optic disc supply is augmented by periodic nocturnal systemic hypotension produced by aggressive antihypertensive drugs and structural axonal crowding at the ONH (due to a disk at risk), with consecutive swelling and ischemia of the disc. These aspects may be mild, subclinical (no visual loss), in general reversible, or may be irreversible (infarction of the ONH), with consecutive NA AION. Because the para-optic branches transport only one-third of the flow of the sPCA, the velocities and RI are relatively normal in NA-AION [50,51,52,53,54].

In a prospective single-center study, Lecler and al. studied which association of imaging techniques should be utilized for the most accurate diagnosis of clinically suspected GCA. They concluded that the best diagnostic algorithm for this purpose consisted in the use of high-resolution 3T magnetic resonance imaging, followed by either extracranial arteries (including temporal and other rami of ECAs) US exam or retinal angiography, before temporal artery biopsy (TAB) [66].

On the one hand, structural optical coherence tomography (OCT) of the ONH offers information about the structural disturbances of the ONH (retinal nerve fiber layer-RNFL-thickness/optic disc edema) of the ONH [10,76].

On the other hand, optical coherence tomography (OCT) angiography may offer additional information concerning the extension of ischemia at the level of ONH (arterial impairments at this level): peripapillary microvessels with eventual microvascular defects (vessel tortuosity, and vessel density reduction) in nonarteritic form of AION [10,76,77].

## 5. Conclusions

US represents a first-line diagnostic investigation for patients presenting with clinical features and biologic data suggesting GCA, taking into account that US has a high sensitivity for identifying the dark halo sign (which represents vessel wall thickening) in the case of a segmental inflammation of large/medium arteries.

For this reason, in our department, US represents a safe and reliable alternative to TAB as a point of care diagnostic tool in the diagnosis of temporal arteritis, or large vessels GCA.

A significant percentage of patients with GCA detected by TAB present ophthalmological features, consisting especially in arteritic form of anterior ischemic optic neuropathy, or central retinal artery thrombotic occlusion.

In acute unilateral CRAO, Color Doppler ultrasonography of the retrobulbar vessels observes a severely diminished or absent blood flow in the CRA of the clinically affected eye, with the normal flow in the homolateral PCAs and OA.

In acute unilateral A-AION, Color Doppler ultrasonography of the retrobulbar vessels reveals a severely diminished or absent blood flow in the PCAs of the clinically affected eye, with normal flow in the homolateral CRA and OA.

Color Doppler US of intraorbital arteries in NA-AION indicates that velocities and RI in PCAs are generally preserved in the clinically affected eye.

## Figures and Tables

**Figure 1 biomedicines-09-01801-f001:**
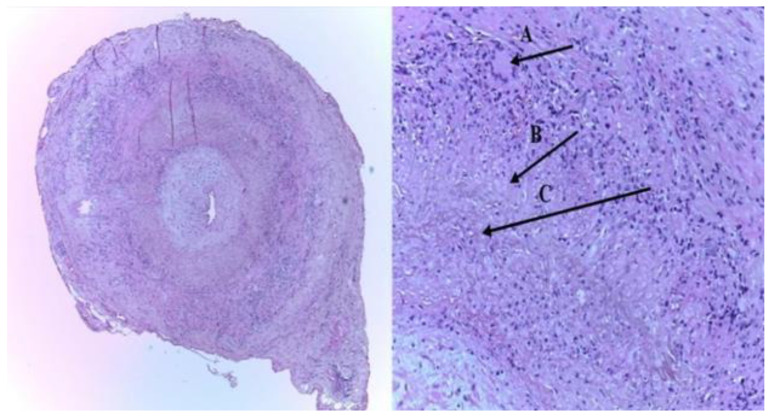
The histopathologic picture of the left superficial temporal artery biopsy (TAB): (A) intimal thickening, and an inflammatory infiltrate with giant cells of the media layer (typical granulomatous inflammation), (B) epithelioid cells, and (C) characteristic internal limiting lamina fragmentation (H&E staining-left-×40; right-×100) [9].

**Figure 2 biomedicines-09-01801-f002:**
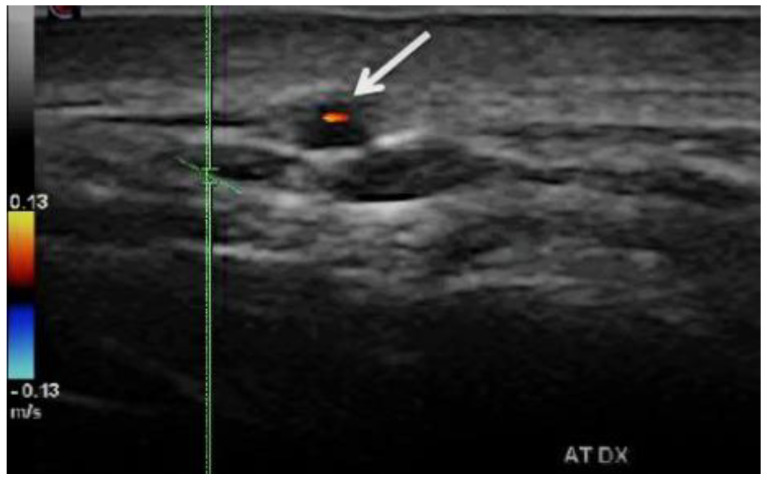
Duplex ultrasound of the right temporal artery−transverse view. The white arrow indicates a “halo” sign (a dark/hypoechoic circumferential wall thickening around the lumen), which represents arterial wall edema [11].

**Figure 3 biomedicines-09-01801-f003:**
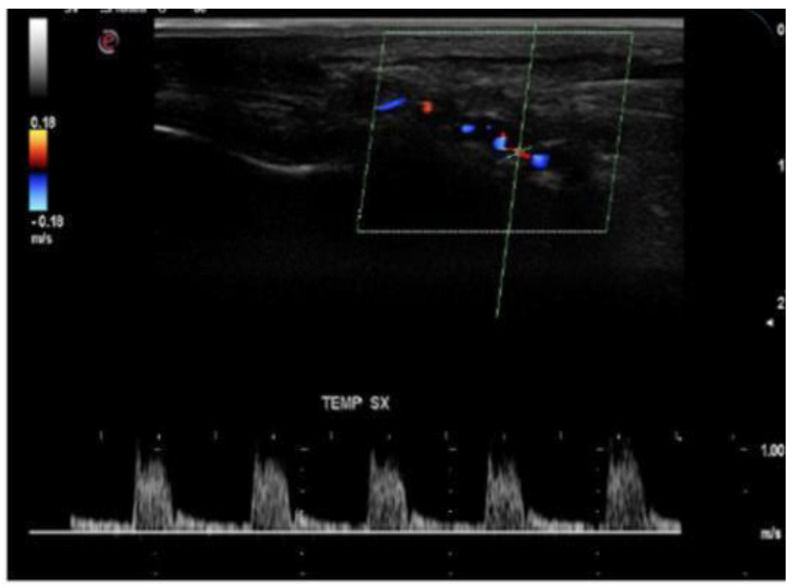
Duplex ultrasound of the right temporal artery−longitudinal view. Indicates a “halo” sign and a stenosis revealed by a turbulent flow and a high PSV in the stenosis area (1 m/s), which is more than twice the PSV in the prestenotic segment of the artery [11].

**Figure 4 biomedicines-09-01801-f004:**
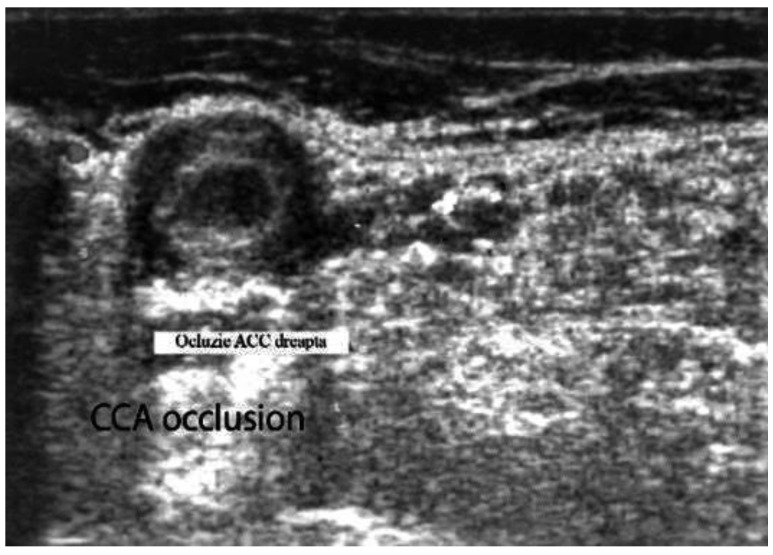
Large vessel GCA. Duplex ultrasound of the right CCA-transverse view. A dark “halo” sign-a hypoechoic circumferential wall thickening around the lumen (which represents arterial wall edema), and occlusion of the artery (the lumen of the vessel is obstructed) [10].

**Figure 5 biomedicines-09-01801-f005:**
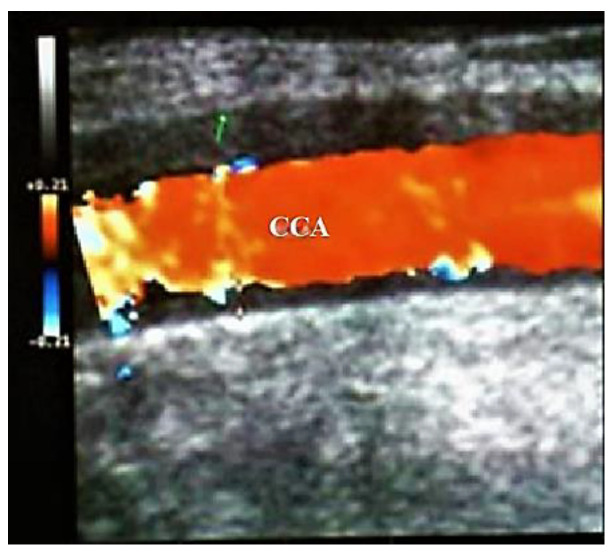
Large vessels GCA. Duplex ultrasound of the right CCA-longitudinal view. The artery presents a dark-hypoechoic circumferential wall thickening (which represents arterial wall edema) [10].

**Figure 6 biomedicines-09-01801-f006:**
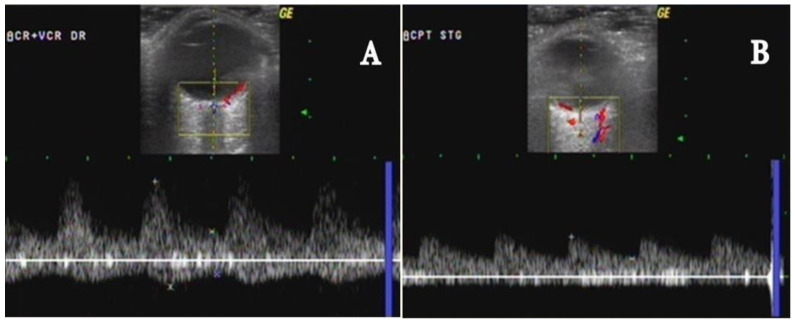
Color Doppler ultrasonography of intraorbital arteries: (**A**) central retinal artery (CRA); (**B**) temporal short posterior ciliary arteries (t-sPCAs)-normal aspects [11].

**Figure 7 biomedicines-09-01801-f007:**
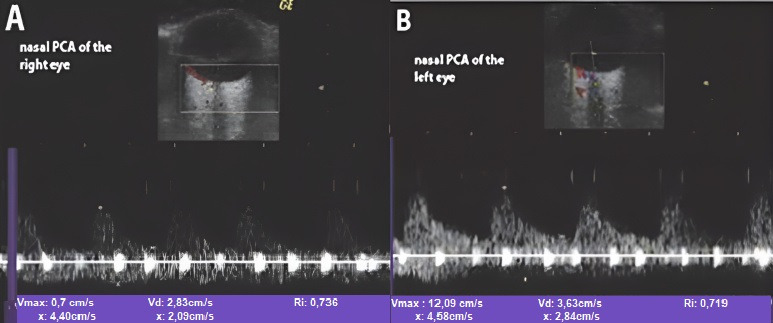
Color Doppler ultrasonography of the PCAs in A-AION: (**A**) severe diminution of EDV in the nasal PCA on the side of clinically affected right eye, and (**B**) significant diminution of EDV in the nasal PCA on the side of clinically nonaffected left eye (Figure 7) [11].

**Figure 8 biomedicines-09-01801-f008:**
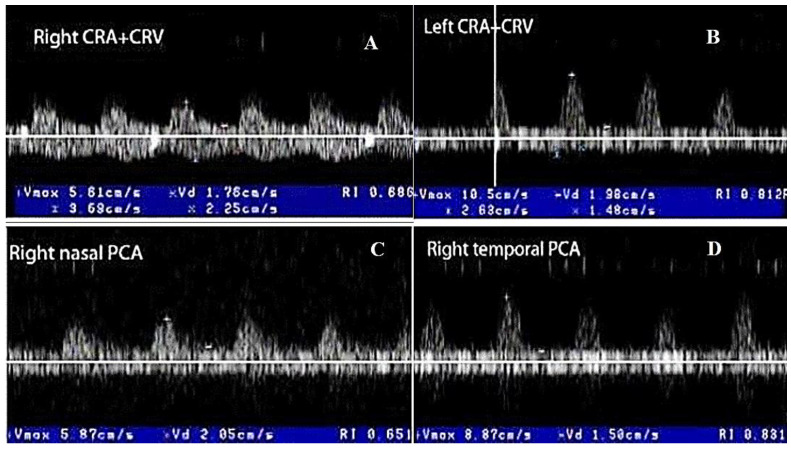
Color Doppler ultrasonography of retrobulbar vessels in the right CRAO produced by GCA: (**A**,**B**) severe diminution of EDV and high RI are observed in both CRAs (less manifested on the left side), even the left eye is clinically nonaffected, and (**C**,**D**) less US abnormalities are noted in the PCAs of both eyes [11].

**Table 1 biomedicines-09-01801-t001:** Normal values of peak systolic velocities (PSV), end-diastolic velocities (EDV), and Resistance Index (RI) in retrobulbar vessels [12,16,60].

Parameter	OA	CRA	PCA (Temporal)	PCA (Nasal)	SOV (Superior Ophthalmic Vein)
PSV (cm/s)	45.3 ± 10.5	17.3 ± 2.6	13.3 ± 3.5	12.4 ± 3.4	10.2 ± 3.8
EDV (cm/s)	11.8 ± 4.3	6.2 ± 2.7	6.4 ± 1.5	5.8 ± 2.5	4.3 ± 2.4
RI	0.74 ± 0.07	0.63 ± 0.09	0.52 ± 0.10	0.53 ± 0.08	

**Table 2 biomedicines-09-01801-t002:** The threshold values of resistance index (RI) in the orbital arteries and the corresponding values of sensitivity (Se), Specificity (Sp), positive predictive value (PPV) and negative predictive value (NPV) in A-AION patients [15].

Arteries	CRA	PCA t	PCA n	OA
Cut-off point	0.67	0.71	0.68	0.81
Se	0.76	0.86	0.86	1
Sp	0.81	0.96	0.93	0.96
PPV	0.51	0.88	0.76	0.89
NPV	0.92	0.96	0.96	1

CRA–central retinal artery; PCAt—temporal posterior ciliar artery; PCAn—nasal posterior ciliar artery; OA—ophthalmic artery.

## Data Availability

First Department of Neurology, “Pius Brînzeu” Emergency County Hospital, Timisoara, Romania.

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
