# Peer review of "Ultrasound Technologies and the Diagnosis of Giant Cell Arteritis"

_biomedicines, 2021, doi:10.3390/biomedicines9121801_

Round 1
Reviewer 1 Report
This is an extensive and detailed review of the use of ultrasonography in the diagnosis of Giant Cell arteritis (GCA). The authors present details about their experience, their technique, and compare it with other diagnostic techniques.
My main criticism of the manuscript as written is that it reads too much as a technical manual of the technique. The portions in which there are comparisons with other techniques, and that provide clinical context are not highlighted. The title could be deceiving, as the “role of ultrasonography” is much less discussed than the ultrasonographic technique.
In addition, the length of the manuscript also makes it a difficult read.
Specific observations:
-The authors call GCA Horton’s disease throughout the manuscript. Although it is correct to explain the eponym at the beginning, the general approach is to avoid using eponyms in general. Especially the ones which are much less well-known than the alternative terms (like in this case GCA).
-Page 2-3 – is such a long description of atherosclerosis needed? Would consider shortening.
-Page 5 – the role of “fast-track” clinics for US use in GCA should be discussed. This includes the protocols proposed for just scanning temporal, axillary, and carotid arteries (Diamantopoulos. Rheumatology 2016;55:66-70).
-A number of statements need to be supported with references. In the conclusions. Page 19, line 784. 25 % of patients with GCA detected by TAB present ophthalmological features. Reference is needed. Similar statements are made throughout the manuscript.
Author Response
Dear Reviewer,
Thank you for your comments and suggestions!
Point 1:
My main criticism of the manuscript as written is that it reads too much as a technical manual of the technique. The portions in which there are comparisons with other techniques, and that provide clinical context are not highlighted. The title could be deceiving, as the “role of ultrasonography” is much less discussed than the ultrasonographic technique. In addition, the length of the manuscript also makes it a difficult read.
Response1: Concerning your main criticism of our manuscript, we wrote a shorter revised version with less technique data. We tried to highlight comparisons with other techniques, and to present US findings in clinical context. We changed the title of our paper in “Ultrasound technologies and the diagnosis of Giant Cell Arteritis”. Point2: The authors call GCA Horton’s disease throughout the manuscript. Although it is correct to explain the eponym at the beginning, the general approach is to avoid using eponyms in general. Especially the ones which are much less well-known than the alternative terms (like in this case GCA). Response 2: We replaced Horton’s disease with GCA throughout our manuscript. Point 3: Page 2-3 – is such a long description of atherosclerosis needed? Would consider shortening.
Response 3: We wrote a shorter description of atherosclerosis
Point4: Page 5 – the role of “fast-track” clinics for US use in GCA should be discussed. This includes the protocols proposed for just scanning temporal, axillary, and carotid arteries (Diamantopoulos. Rheumatology 2016;55:66-70). Response4: We discussed the role of “fast-track” clinics for US use in GCA (Diamantopoulos 2016) Point5: A number of statements need to be supported with references. In the conclusions. Page 19, line 784. 25 % of patients with GCA detected by TAB present ophthalmological features. Reference is needed. Similar statements are made throughout the manuscript. Response5: We checked carefully our manuscript, so, in the modified version different statements were supported with references.
Reviewer 2 Report
It is a good work.
The authors present a comprehensive analysis of imaging techniques in giant cell arteritis , focusing on the role of ultrasonography and listing the common and uncommon US abnormalities in giant cell arteritis.
Furthermore, the authors provided the definition of US indexes useful to evaluate vessels involvement.
The work is well presented and supply a complete summary about diagnostic tools in giant cell arteritis.
Could be improved with adding the Method section (with research methods, articles included and those excluded, etc...) and clarifying the objective of the study.
Author Response
Dear Reviewer,
Thank you for your supportive comments and suggestions!
We wrote a shorter revised version with less technique data. We tried to highlight comparisons with other techniques, and to present US findings in clinical context. We changed the title of our paper in “Ultrasound technologies and the diagnosis of Giant Cell Arteritis”.
Point1: Could be improved with adding the Method section (with research methods, articles included and those excluded, etc...) and clarifying the objective of the study.
Response1: Early and accurate diagnosis of giant cell arteritis (GCA) is essential to prevent the most serious complications that are ophthalmological ischemias. Diagnostic confirmation of GCA remains challenging. There is a wide spectrum of clinical features in GCA, sometimes with nonspecific clinical or biological signs. The purpose of our review was to identify different imaging modalities (especially US) that should be used to obtain the best diagnostic performance for the non-invasive diagnosis of GCA. For this reason, we searched different studies with GCA patients concerning single US or a combination of imaging modalities used to diagnose GCA.
Round 2
Reviewer 1 Report
The authors have addressed comments, The manuscript is shortened which makes it more readable